# New Vaccine Introduction in Middle-Income Countries Across the Middle East and North Africa—Progress and Challenges

**DOI:** 10.3390/vaccines13080860

**Published:** 2025-08-14

**Authors:** Chrissy Bishop, Deeksha Parashar, Diana Kizza, Motuma Abeshu, Miloud Kaddar, Abdallah Bchir, Atef El Maghraby, Hannah Schirrmacher, Zicheng Wang, Ulla Griffiths, Shahira Malm, Sowmya Kadandale, Saadia Farrukh

**Affiliations:** 1Triangulate Health Ltd., Doncaster DN11 9QU, UK; 2UNICEF MENA Regional Office, Abdulqader Al-Abed Street, Building No.15 Tla’a Al-Ali, Amman 1723, Jordan; 3MK International Consulting, 75000 Paris, France; 4Department of Community Medicine, Faculty of Medicine, Monastir Medical School, University Hospital Monastir, Monastir 5000, Tunisia; 5Independent Researcher, Cairo 11435, Egypt; 6UNICEF Supply Division, Oceanvej 10-12, 2150 Copenhagen, Denmark; 7UNICEF Head Quarters, New York, NY 10017, USA

**Keywords:** new vaccine introduction, middle-income countries, Middle East and North Africa, human papillomavirus vaccine, pneumococcal conjugate vaccine, rotavirus vaccine, immunisation barriers and facilitators

## Abstract

**Background/Objectives**: The middle-income countries (MICs) in the Middle East and North Africa (MENA) region face multifaceted challenges—including fiscal constraints, conflict, and vaccine hesitancy—that impede the timely introduction of critical vaccines. This study examines the status, barriers, and facilitators to introducing three critical vaccines—human papillomavirus vaccine (HPV), pneumococcal conjugate vaccine (PCV), and rotavirus vaccine (RV)—across seven MENA MICs, to identify actionable solutions to enhance vaccine uptake and immunisation coverage. **Methods**: Using the READ methodology (ready materials, extract, analyse, and distil data), this review systematically analysed policy documents, reports, and the literature on the introduction of HPV, PCV, and RV vaccines in seven MENA MICs. A data extraction framework was designed to capture the status of vaccine introduction and barriers and facilitators to introduction. Findings and data gaps were validated with stakeholder consultations. **Results**: Of the seven study countries, progress in introducing PCV and RV has been uneven across the region (five countries have introduced PCV, four have introduced RV, and only a single country has introduced HPV at time of writing), hindered by vaccine hesitancy, fiscal challenges, and insufficient epidemiological data. Morocco is the only country to introduce all three vaccines, while Egypt has yet to introduce any. Other common barriers include the impact of conflict and displacement on healthcare infrastructure, delayed introduction due to the 2020 COVID-19 pandemic, and limited local production facilities and regional cooperation. In addition, not all countries eligible for Gavi MICs support have applied. These findings provide a roadmap for policymakers to accelerate equitable vaccine introduction in the MENA region. **Conclusions**: Targeted efforts, such as addressing fiscal constraints, improving local manufacturing, tackling gender barriers, and fostering public trust, paired with regional collaboration, can help bridge gaps and ensure no community is left behind in preventing vaccine-preventable diseases.

## 1. Introduction

The challenges associated with vaccine introduction are particularly pronounced in the Middle East and North Africa (MENA) region’s middle-income countries (MICs). Unlike low-income countries, many MICs have been ineligible for support from the Gavi, the Vaccine Alliance, which has been instrumental in accelerating vaccine introductions in low-resource settings [1]. Without this support, MICs face significant barriers, including the high cost of vaccines, fiscal constraints, and disruptions to healthcare delivery caused by conflict and displacement [2]. The pace of vaccine introduction in MENA’s MICs has lagged compared to that of high-income countries and Gavi-supported low-income countries.

In response to these challenges, Gavi has developed a MICs approach to support vaccine introduction under the Gavi 5.0 strategic period (until 2025). This focuses on preventing declines in vaccine coverage in former Gavi-eligible countries and promoting the sustainable introduction of key vaccines in both former and select never-Gavi-eligible countries [3]. At the time of writing (2024), eight MICs across the MENA region—Algeria, Egypt, Iran, Jordan, Lebanon, Morocco, The State of Palestine, and Tunisia—were eligible for this support [3]. The State of Palestine was not included in this study, as resources were redirected to urgent priorities following the outbreak of war in October 2023.

The MICs approach aims to prevent declines in vaccine coverage in former and never-Gavi-eligible countries and support the sustainable introduction of essential vaccines in both former and select never-Gavi-eligible countries [3]. This presents a timely opportunity to accelerate the introduction of critical vaccines for addressing illnesses affecting children and women, and improve immunisation coverage across the region [3].

New vaccines address preventable diseases that disproportionately affect women and children in the region. The human papillomavirus (HPV) vaccine can help reduce the burden of cervical cancer, while the pneumococcal conjugate vaccine (PCV) and rotavirus vaccine (RV) target pneumonia and severe diarrhoea—two leading causes of childhood mortality, especially in low-and-middle-income countries [4,5]. Despite the availability of these vaccines and their proven benefits, their rollout in the MENA region has been slow and uneven, leaving large segments of the population at risk [6,7,8].

Given the persistent challenges and the new opportunities presented under the Gavi MICs approach, this review seeks to assess the status of new vaccine introductions (NVIs), specifically HPV, PCV, and RV—focus vaccines of Gavi MICs 5.0 [3]. This study aims to analyse the political, economic, and health system barriers that have hindered the timely adoption of these vaccines, while also identifying the facilitators that have enabled successful implementation in certain contexts. By examining these dynamics, this review aims to provide evidence-based insights to inform policies and strategies that can strengthen immunisation systems and promote equitable vaccine access across the region.

## 2. Materials and Methods

The present literature review used the ready materials; extract data; analyse data; and distil data approach, or the READ approach, to systematically analyse policy documents, reports, and the related literature concerning the introduction of HPV, PCV, and RV vaccines in the seven focus countries [9]. The READ method for document analysis is a structured process used to gather and extract information from documents, particularly within the scope of health policy research at various levels, including global, national, and local contexts [9]. The goal was to evaluate the status of new vaccine introductions and associated influencing factors. The details of the methodology and its rationale for selection for this study are given in the appendix.

### 2.1. Ready Materials (R)

The review aimed to capture governmental policies, documents outlining financing mechanisms, technical advisory group recommendations, and local manufacturing capabilities related to these vaccines. Inclusion criteria were materials published within the last decade and directly relevant to the introduction status of HPV, PCV, and RV vaccines in the seven MICs. Documents that lacked relevance or specificity were excluded. Search terms such as “HPV vaccine introduction MENA”, “rotavirus vaccine policy Jordan”, “PCV supply Algeria” were iteratively refined based on preliminary findings and emerging themes, ensuring thorough and focused document identification. The complete search strategy is provided in the appendix.

### 2.2. Extract Data (E)

Data were systematically extracted using a bespoke data extraction tool to assess the introduction of HPV, PCV, and RV vaccines in seven MICs. Based on the extraction tool, and the categories of information arising, a vaccine readiness framework (VRF) was developed to assess vaccine preparedness across five domains: epidemiology, health system capacities, financial resources, political economy, and local manufacturing capabilities (Figure 1). The vaccine readiness framework was further refined based on insights from two rounds of expert consultations to capture the following supplementary domains: socio-political context, immunisation expenditure, vaccine manufacturing capabilities, and political and economic situation.

### 2.3. Expert Consultation

In addition to document analysis, we conducted two rounds of expert consultations:•UNICEF MENARO and UNICEF country office consultations: We engaged with UNICEF Middle East and North Africa Regional Office (MENARO) and country offices to gather region-specific insights and data at the country level, supplementing our findings.•Expert consultations: Consultations with experts in immunisation, public health, and vaccine policy were conducted to identify barriers and facilitators to vaccine introduction. These included experts from the MENA region, UNICEF regional office, and UNICEF country offices. In addition, we also gathered inputs and validated findings from an expert Steering Committee, consisting of members from the World Health Organisation, UNICEF Supply Division (SD), and Gavi, the Vaccine Alliance. These consultations provided contextual insights into challenges such as political will, financing, and public awareness that may not have been fully captured in the documents.

Data was extracted by country and theme to facilitate comparative analysis. Documents were reviewed by two researchers to ensure thoroughness, and extracted data were organised by country to facilitate comparisons and identify data gap commonalities.

### 2.4. Analyse Data (A)

Data were analysed to identify commonalities across the seven countries in vaccine introduction strategies and gaps in data availability.

### 2.5. Distil Data (D)

The findings were synthesised into key epidemiological, financial, procurement, and supply chain insights, as well as outlining the main barriers and facilitators for each country, evidence gaps, practical recommendations, and areas for further research. Insights from expert consultations were integrated to validate the findings and provide additional depth and contextualisation.

## 3. Results

### 3.1. Epidemiology of Vaccine-Preventable Diseases

Table 1 summarises country-specific estimates for rotavirus-positive acute gastroenteritis, pneumonia and severe pneumonia in children, and cervical cancer in women. For rotavirus, only period prevalence data among children admitted to hospital with acute gastroenteritis were available from various hospital surveillance studies; no age-standardised data were found. Period prevalence varied widely, from 6% in Jordan (post-vaccine) to 30% in Tunisia [10,11]. In Algeria, no rotavirus prevalence data were available. The WHO’s Rotavirus Laboratory Network monitors rotavirus and associated diarrhoea prevalence, but MENA countries are mostly excluded [12]. The burden of rotavirus in these countries thus remains unclear.

The latest data on pneumococcal disease come from a global study by Wahl et al. [13], using WHO and country-specific estimates of pneumonia and meningitis mortality and morbidity from 2000 to 2015. Pneumonia incidence was 2% in all countries apart from Egypt and Lebanon, which had an incidence of 1%.

The latest data on cervical cancer in women come from the HPV Information Centre, which derives data from a systematic review and meta-analysis of the published literature and official reports by the WHO, the United Nations, The World Bank, and IARC’s Globocan and Cancer Incidence data until 2022. This source reports age-standardized cervical cancer incidence for year 2020 [14]. Morocco had the highest incidence of cervical cancer of 0.01%, with Iran having the lowest incidence of 0.002%. It is estimated that 95% of cervical cancers are caused by HPV infection [14].

**Table 1 vaccines-13-00860-t001:** Burden of vaccine-preventable diseases.

	Algeria	Egypt	Iran	Jordan	Lebanon	Morocco	Tunisia
Period prevalence of rotavirus-attributable gastroenteritis/severe, diarrhoea in children under five years of age in hospital settings with acute gastroenteritis *	Unknown	24% [15]	~60% [16]	6% [10]	17% [17]	24% [18]	30.3% [19]
Age-standardised one year incidence of pneumonia (severe) in children under five years of age, 2015 [13]	2% (0.8%)	1% (0.4%)	2% (0.9%)	2% (0.9%)	1% (0.4%)	2% (0.8%)	2% (0.8%)
Age-standardised one year incidence rate of cervical cancer per 100 k women (%), 2020 [14]	7.9 (0.008%)	2.61 (0.003%)	2.33 (0.002%)	2.91 (0.003%)	3.4 (0.003%)	10.4 (0.01%)	4.56 (0.005%)

* Data reporting period for the hospital prevalence estimates were as follows; Egypt October 2015 to September 2017, Iran May 2006 to April 2007, Jordan December 2018 to August 2019, Lebanon January to December 2018, Morocco December 2011 to January 2013, Tunisia April 2009 to March 2011.

### 3.2. The State of New Vaccine Introductions (NVI)

Across the MENA region, there has been steady progress in integrating PCV and RV into the NIPs of MICs, with countries like Morocco, Iran, and Lebanon having introduced both [8,20]. However, the introduction of HPV has been much slower, with only Morocco having implemented it so far [21].

The recommendations for new vaccine introductions are made by the NITAGs of respective countries [22]. These advisory groups are tasked with evaluating evidence and making recommendations for introduction of new vaccines, optimisation of immunisation schedules, and advising on other immunisation strategies based on local epidemiology, public health significance, and global standards [22]. However, detailed NITAG recommendations or the evidence used to support these decisions are not publicly accessible for most MENA countries. Limited information is available in literature, with one example being the 2015 NITAG recommendation in Jordan, which endorsed the introduction of PCV and RV into the National Immunisation Program [23,24].

#### 3.2.1. Rotavirus (RV) Vaccine

The introduction of RV across the MENA region shows significant variation (Table 2). Morocco and Lebanon have made substantial progress in integrating RV into their NIPs, with Morocco achieving a coverage rate of 98% by 2023 [25]. Lebanon introduced RV in 2022, reaching 40% coverage by 2023 [25]. In Jordan, RV was introduced into the National Immunisation Program in 2015 following a NITAG recommendation and has reached 96% coverage [25]. Iran commenced phased RV introduction in 2024, starting in select provinces with plans for nationwide expansion [26]. As of 3 December 2024, President Raisi announced that an adequate supply of the rotavirus vaccine had been distributed nationwide, marking the full-scale rollout [27]. In contrast, Egypt, Tunisia, and Algeria have yet to consider including RV in their NIPs.

Procurement for RV is largely managed through UNICEF or self-procurement mechanisms, with government funds being the primary financing source in countries with National Immunisation Program inclusion. In Egypt, RV is only available through private channels, requiring individuals to pay out-of-pocket.

Gavi MICs support for the rotavirus vaccine varies across MENA countries. Egypt and Tunisia, despite being eligible, have not applied for Gavi MICs support, delaying potential integration into their NIPs. Iran has applied for Gavi MICs support to aid its phased introduction, whereas Jordan, Lebanon, and Morocco introduced RV before the MICs strategy, relying entirely on government funds for financing.

**Table 2 vaccines-13-00860-t002:** RV introduction status (as of October 2024).

RV	Indicator	Algeria	Egypt	Iran	Jordan	Lebanon	Morocco	Tunisia
Vaccine supply	Vaccine in National Immunisation Program (NIP) [20]	No	No	Yes	Yes	Yes	Yes	No
	Year of introduction in NIP [20]	N.A.	N.A.	2024 (in select provinces)	2015	2022	2010	N.A.
	Vaccine in private channels	No	Yes [28]	Yes [29]	Yes [30]	Yes [30]	Yes [26]	Yes [30]
	Vaccine/valence [31]	N.A.	Unknown	Rotasiil	Rotarix; Rotateq	Rotarix	Rotasiil	Unknown
Procurement and financing	Source of procurement [26,29]	N.A.	Imported [26,29]	UNICEF	Self-procured	UNICEF	Self-procured	Imported
	Source of financing [26,29,32]	N.A.	Out-of-pocket for the individual	Government funds	Government funds	Government funds and Gavi support	Government funds	Out-of-pocket for the individual
Gavi MICs support	Eligibility, Status of application [33]	Not eligible	Eligible, Not applied	Eligible, applied	Not eligible, (vaccine introduced before MICs strategy)	Not eligible (vaccine introduced before MICs strategy)	Not eligible (vaccine introduced before MICs strategy)	Eligible, not applied
Vaccine coverage	Coverage % (2023, WUENIC) [25]	N.A.	N.A.	Recently introduced	96%	40%	98%	N.A.

*N.A.: not applicable.*

#### 3.2.2. Pneumococcal Conjugate Vaccine (PCV)

PCV uptake in children under 5 years of age has seen widespread adoption across the MENA region, with varying levels of success (Table 3). Morocco and Tunisia have demonstrated leadership in PCV introduction, achieving high coverage rates of 98% and 97%, respectively, by 2023 [34]. Notably, Morocco recently signed a strategic deal with Walvax, a Chinese biopharmaceutical company, to secure a cost-effective PCV13 supply, further strengthening its immunisation efforts [35]. Algeria also reported strong coverage of 89% following its introduction of PCV13 into the National Immunisation Program in 2016, while Lebanon reports moderate coverage (65%) [34]. However, Egypt and Jordan have yet to integrate PCV into their National Immunisation Program [8]. In Jordan, the NITAG recommended PCV introduction as early as 2015, but fiscal challenges have delayed implementation [24]. The country is currently working with Gavi, UNICEF, and WHO to introduce PCV13 in 2025 [29]. The phased introduction of PCV10 in Iran began in 2024, starting with select provinces and with plans for nationwide rollout, reflecting a prioritisation of cost-effective vaccine procurement despite financial limitations [36].

Countries with PCV in their NIPs rely on a mix of UNICEF procurement (e.g., Morocco and Lebanon) or direct self-procurement (e.g., Tunisia and Algeria) [26,29,37]. Financing is primarily supported by government funds in countries with National Immunisation Program inclusion, while Egypt and Jordan rely on private payments for PCV access [26].

Countries that introduced PCV before Gavi’s MICs strategy, such as Morocco, Lebanon, and Tunisia, are ineligible for Gavi MICs support for new vaccine introductions. Iran and Jordan have applied for Gavi MICs support to introduce PCV and its application has been approved by Gavi’s Internal Review Committee, as per recent documents dated July 2024 [33]. Although Egypt is eligible, it has not applied for support, thereby continuing its dependence on private market access [26,38].

**Table 3 vaccines-13-00860-t003:** PCV introduction status (as of October 2024).

PCV	Indicator	Algeria	Egypt	Iran	Jordan	Lebanon	Morocco	Tunisia
Vaccine supply	Vaccine in National Immunisation Program (NIP) [8]	Yes	No	Yes	No (Recommended by NITAG in 2015) [24]	Yes	Yes	Yes
	Year of introduction in NIP [8]	2016	N.A.	2024 (in select provinces)	N.A.	2010	2010	2019
	Vaccine in private channels	No [26]	Yes [39]	Yes [26]	Yes [26]	Yes [26]	Yes [26]	No [26]
	Vaccine name/valence	PCV 13 [40]	Unknown	PCV 10 (NIP) [40]	PCV 10 (private channels)	PCV 10 (in NIP, transitioned from PCV 13) [26,40]	PCV 10 and PCV 13 [40]	PCV 10 [40]
Procurement and financing	Source of procurement	International market from Pfizer via the national procurement agency, Pharmacie Centrale des Hopitaux [26]	Imported for distribution in the private market [26]	UNICEF [36]	Imported for private distribution [26]	UNICEF SD [26]	Self-procured [26,37]	Self-procured (via Pharmacie Centrale de Tunisie) [26,37]
	Source of financing [26,29]	Government funds	Out-of-pocket for the individual or private insurance	Government funds	Out-of-pocket for the individual or private insurance	Gavi + Government funds	Government funds	Government funds
GAVI MICs support	Eligibility, Status of application [33]	Not eligible	Eligible, has not applied	Eligible, applied	Eligible, applied	Not eligible (vaccine introduced before MICs support)	Not eligible (vaccine introduced before MICs support)	Not eligible (vaccine introduced before MICs support)
Vaccine coverage	Coverage % (2023, WUENIC) [5]	89% (2023)	N.A.	NA—recently introduced	N.A.	65% (2023)	98% (2023)	97% (2023)

*N.A.: not applicable.*

#### 3.2.3. HPV Vaccine

The landscape of HPV vaccine introduction across the seven MENA countries reflects a diverse range of challenges and progress (Table 4). Morocco has successfully introduced the HPV vaccine into their National Immunisation Program, although the coverage remains unknown, and Tunisia is initiating its rollout in 2025 [38,41]. Conversely, Lebanon, Egypt, and Jordan have yet to introduce the HPV vaccine into their NIPs. HPV, however, is available through private channels. The affordability of the vaccine if bought from the private sector remains a significant barrier in these countries, with prices ranging from USD 30 and USD 160 per dose in Egypt and Jordan, respectively, limiting financial accessibility [42]. Similarly, Algeria and Iran have not yet introduced the HPV vaccine into their NIPs [33]. Global supply shortages have compounded the delays in HPV vaccine rollouts across several countries, with many low- and middle-income nations experiencing postponed introductions [43].

Morocco procures HPV via UNICEF and funds the programme entirely through the government, while Tunisia plans to procure—and has recently introduced—HPV vaccines through UNICEF with support from Gavi (under ‘vaccine catalytic financing’ where Gavi may provide vaccine financing for half the first birth cohort) and government funds [44]. In countries where the HPV vaccine is only available in private markets, individuals bear the full cost out-of-pocket, leading to significant inequities in access. Tunisia is the only country in the region that has applied for Gavi support for HPV introduction. Other Gavi-eligible countries in the region have not applied, leaving them reliant on private channels and individual financing [33].

Awareness of the HPV vaccine, its role in preventing cervical cancer, and the link between human papillomavirus and the disease remains low across much of the MENA region, posing a significant barrier to uptake and public acceptance. In Algeria, only 46.2% of students had heard of the HPV vaccine, with just 21.5% expressing willingness to receive it in one study [45]. A study in Egypt revealed that a significant proportion of physicians, including 41.2% Ob-Gyn specialists and 37.6% consultants, had poor-to-fair knowledge of HPV and cervical cancer screening, with only 45% having ever prescribed the HPV vaccine, highlighting gaps in knowledge and practice influenced by age, professional level, experience, and workplace setting [46]. In Jordan and Lebanon, limited public health campaigns have contributed to low levels of awareness, particularly in rural areas, where understanding of the vaccine’s role in preventing cervical cancer is sparse [47,48]. Surveys indicate that despite HPV introduction in Morocco in 2022, substantial efforts are needed to raise public understanding of the vaccine in Morocco to achieve higher acceptance and coverage rates [49]. Across all studies researching attitudes towards HPV, awareness and vaccine hesitancy is reported as a major barrier to uptake [50].

**Table 4 vaccines-13-00860-t004:** HPV introduction status (as of October 2024).

HPV	Indicator	Algeria	Egypt	Iran	Jordan	Lebanon	Morocco	Tunisia
Vaccine supply	Vaccine in National Immunisation Program (NIP) [21]	No	No	No	No	No	Yes	No, planned introduction in 2025 [41,51]
	Year of introduction in NIP [21]	N.A.	N.A.	N.A.	N.A.	N.A.	2022	Planned for 2025
	Vaccine in private channels	No [26]	Yes [42]	Yes [52]	Yes [48]	Yes [47,53]	Yes [54]	No (available for several years but not anymore) []
	Vaccine name/valence	N.A.	Gardasil 4, Cervarix	Gardasil 4 [55]	Gardasil 4 [30]	Gardasil, Cervarix [56]	Gardasil 4	Bivalent (planned) []
	Target group (if introduced in NIP)	N.A.	N.A.	N.A.	N.A.	N.A.	Girls aged 11 [49]	Girls aged 12 who are currently in school (out of school girls missed) [41]
Procurement and financing	Source of procurement	N.A.	Imported for distribution through private channels [26]	Imported for distribution through private channels [26]	Imported for distribution through private channels [26]	Imported for distribution through private channels [26]	UNICEF SD [44]	UNICEF SD []
	Source of financing	N.A.	Out-of-pocket for the individual [29]	Out-of-pocket for the individual [52]	Out-of-pocket for the individual [29]	Out-of-pocket for the individual [29]	Government funds [44]	Government funds (+ Gavi for half of the first birth cohort) []
Gavi MICs support	Eligibility/status of application [33]	Not eligible	Eligible, has not applied	Eligible, has not applied	Eligible, has not applied	Eligible, has not applied	Eligible, has not applied	Eligible, applied
Vaccine coverage	Coverage %	N.A.	N.A.	N.A.	N.A.	N.A.	55% (Ministry of Health estimate, 2022) [34]	N.A.

*N.A.: not applicable.*

### 3.3. Supplementary Findings

#### 3.3.1. Immunisation Financing

##### Government Expenditure on Vaccines Used in Routine Immunisation

Data sourced from Joint Reporting Form (JRF) submissions reveal significant variation in government expenditure on vaccines used in routine immunisation across the seven MENA countries, with both similarities and differences in the approach to public financing (Figure 2). Algeria, Jordan, and Iran demonstrate increases in their respective vaccine budgets. For example, Algeria’s vaccine budget surged from DZD 24 billion (~USD 169 million) in 2022 to 36 billion DZD (~USD 266 million) in 2023 [57]. Similarly, Jordan’s government vaccine expenditure rose from USD 20 million in 2013 to USD 35 million in 2022, with 82% of costs covered by the government. Iran’s expenditure also increased from USD 21 million in 2012 to USD 82 million in 2022, with the government financing 96% of vaccine costs [57]. These figures underline the strong public sector commitment to immunisation financing in these countries.

In contrast, Morocco and Tunisia display more modest increases in government spending. Morocco’s spending remained relatively stable, rising from USD 52 million in 2013 to USD 56 million in 2022 [57]. Tunisia’s vaccine expenditure grew from USD 5.1 million in 2012 to USD 14 million in 2023 [57]. Tunisia also integrates increasing private sector funds, especially in urban areas [30]. Egypt has shown significant growth in immunisation funding, with government expenditure increasing from EGP 32 million (~USD 50 million) in 2012 to EGP 1.6 billion (~USD 95 million) in 2019 [57].

Lebanon showed an increasing trend in government expenditure, from USD 3 million in 2012 to USD 6 million in 2021. Lebanon’s government expenditure on vaccines in 2021 was lower than the total spending on vaccines from all funding sources, which amounted to USD 21 million [57]. This is not a reflection of Lebanon’s commitment, but rather the fiscal crisis and diversion of funds towards regional war [26].

A study evaluating the per capita spend on immunisation in the MENA region, published in 2021 found that, between 2010 and 2019, most countries showed increases in vaccination expenditure per capita. Algeria, Iran, Jordan, Lebanon, and Morocco showed an increase, whereas Egypt showed a decrease in vaccine expenditure per capita between the time period. No clear trend was reported for Tunisia [58]. This study also found significant variations in per capita vaccine spending across the MENA region (Table 4). For instance, Jordan had the highest per capita expenditure at USD 2.93 in 2019, exceeding the lower-middle-income countries (LMICs) average. In contrast, Tunisia’s per capita spending was the lowest, at TND 0.45 in 2017, falling below the LMIC average of TND 0.55 in 2011 and TND 1.01 in 2014 [58]. Furthermore, evidence suggests that MICs in the MENA region often pay higher prices for vaccines than their counterparts in other regions, exacerbating inequities in spending [32]. Several factors contribute to this, including limited access to pooled procurement mechanisms, unpredictability in the timeliness of cash disbursements for vaccine procurement, and inadequate global vaccine supply to meet demand [59].

##### Private Sector Involvement in Immunisation

The literature and data on private sector involvement in immunisation across the MENA region are limited. A survey conducted by the UNICEF Regional Office in April 2020, which included responses from 17 MENA countries, revealed this variation in private sector roles and coverage. In Tunisia and Lebanon, up to 40% and 50% of routine vaccinations, respectively, are administered through private providers, making it the highest proportion in the region [30,60]. Conversely, Algeria and Iran report minimal private sector involvement, with private providers contributing less than 1% of vaccinations. Similarly, in Jordan, private providers account for less than 2% of routine vaccinations, with their role being more prominent in administering non-scheduled vaccines, such as hepatitis A and varicella [30].

#### 3.3.2. Local Vaccine Manufacturing Capabilities

Domestic manufacturing is of growing focus in the MENA region and increases sustainability of vaccine supply due to less reliance on imports [61]. Global initiatives, such as Gavi’s African Vaccine Manufacturing Accelerator (AVMA) and the PAVM, offer guidance and opportunities for MENA countries to scale up local production and reduce dependence on external suppliers [61]. Levels of domestic manufacturing preparedness vary significantly however (Table 5). Despite having some local production infrastructure in the MENA region, these countries continue to rely heavily on imported vaccine components, which remains a challenge particularly for new vaccines such as HPV or PCV. Among the factors limiting expansion of local vaccine production is the lack of strategies, investment, innovation capacity, partnership, and support from National Regulatory Authorities (NRA) to accelerate relevant processes [61,62]. Egypt’s National Regulatory Authority (NRA) stands out as the only one in the region to achieve maturity level 3, which is critical for accelerating vaccine production processes and regulatory approvals [63].

Egypt leads the region in terms of diversity of vaccine manufacturing portfolio with Vacsera handling both DS and/or F&F for 21 different vaccines (Table 6). These include PCV and HPV vaccines, albeit fill and finish (F&F) only. Algeria and Egypt’s manufacturers have the highest sales volume of pharmaceuticals in the whole MENA region (a total of 18 countries), at 77.5% and 63.9%, respectively [64]. Iran’s Razi Serum and Pasteur Institutes are also producing key vaccines such as MMR and polio, while newer private sector initiatives are expanding capabilities [62,64,65]. Tunisia and Algeria also have well-established vaccine production capabilities through their Pasteur Institutes, which manage both DS production and F&F operations for vaccines like BCG and rabies [61,62]. Morocco focuses primarily on F&F processes, importing the necessary components for vaccine assembly. Jordan and Lebanon lack significant domestic vaccine production infrastructure, relying entirely on imports [26,62].

**Table 6 vaccines-13-00860-t006:** Vaccine manufacturing capabilities in the MENA region [62].

Country	Major Vaccine Manufacturers	Vaccines Manufactured	Existing Capabilities
Algeria	Institut Pasteur d’Algérie (Founded in 1894) Saidal (Founded in 1982)	Anti-rabies	DS, F&F
Egypt	Egy Vac (Vacsera, founded in 1897) Minapharm (Founded in 1958) Biogeneric pharma	F&F—cholera, Covid-19, DT, DTP, tetanus, typhoid Import for distribution—Hib, hepatitis A/B, HPV, influenza, BCG, IPV, meningitis, measles–mumps–rubella (MMR), OPV, Penta, PCV, rabies, rotavirus, varicella, yellow fever	F&F BGM signed an MoU with Sanofi
Iran	Razi Institute, Karaj, Iran Institute Pasteur of Iran, Tehran, Iran Shifa Pharmed Industrial Co., Tehran, Iran CinnaGen Co., Alborz Province, Iran	BCG, cholera, recombinant hepatitis B, measles, polio, MMR, trivalent vaccines, and divalent	DS
Jordan	No information available	No information available	No information available
Lebanon	No information available	No information available	No information available
Morocco	Institut Pasteur du Maroc (Founded in 1967)	Import for distribution—BCG, influenza, rabies, tetanus, typhoid, yellow fever	Importing for distribution
Tunisia	Institut Pasteur de Tunis (Founded in 1893)	BCG	DS, F&F

#### 3.3.3. Political and Economic Context

Political and economic instability across the MENA region, exacerbated by the recent wars and conflicts, have impacted new vaccine introductions and their sustainability. Countries like Egypt, heavily reliant on tourism and Suez Canal revenues, face severe economic strain. Egypt’s economy has been facing further external shocks from the pandemic, war, and the recent Red Sea attacks [66,67]. A UNDP assessment projects a GDP decline of up to 3% for Egypt in 2023–24, with potential revenue losses of up to USD 13.7 billion [67]. Iran continues to struggle under international sanctions, high inflation (42% in 2022), and currency devaluation, which have slowed the rollout of new vaccines [68]. The re-imposition of U.S. sanctions and supply chain disruptions from the Russia–Ukraine war have further restricted Iran’s economic recovery [68]. Jordan, despite significant foreign assistance from the U.S (USD $1.45 billion annually through 2029), faces high levels of debt (114% of gross domestic product (GDP)), which may have slowed vaccine programme expansion. Algeria, despite a 4.2% increase in its economic growth rate in 2023, continues to struggle with high inflation (9.3%) and a budget deficit (10.2% of the GDP in 2023) [69]. Lebanon remains in economic crisis, with triple-digit inflation (231% in 2023) and a sovereign debt-to-GDP ratio of 179%, leading to severe economic contraction [70]. In contrast, Morocco has demonstrated greater resilience, introducing all three vaccines into its national immunisation programme. Its economy grew by 3.02% in 2023, driven by agriculture, tourism, and stable domestic demand [68,71].

The COVID-19 pandemic also caused disruptions in the introduction of new vaccines. In Jordan, resource constraints during the pandemic led to the de-prioritisation of introducing PCV into the national immunisation schedule. Additionally, a global study found that during the pandemic period, worsening HPV vaccine coverage and delayed introductions of national HPV vaccination programmes were observed [24,72].

The MICs of the MENA region face further challenges supporting migrants, refugees and displaced persons. By end of 2022, they were hosting about 2.4 million refugees, 251,800 asylum-seekers, 370,300 stateless persons and 12.6–16.2 million internally displaced people [73,74]. Iran counted 2.6 million undocumented Afghans (with another 500,000 likely unrecorded); Lebanon sheltered 1.5 million Syrians, while Jordan registered 730,000 refugees, four-fifths living outside camps, and continuous flows through Morocco, Algeria, and Tunisia added further volatility [75,76,77].

Children make up roughly 40–50% of these displaced populations, yet a recent systematic review which included MENA countries indicates only 36% of migrant children are fully vaccinated according to national schedules, with coverage for key vaccines such as DTP3 ranging from 59.7% to 76.6% and MCV2 from 25.4% to 85.6% [78]. These figures are far below those for host populations, highlighting disparities in access to vaccination services [78]. Studies show that some barriers to immunisation in displaced populations include systemic issues such as vaccine shortages, administrative challenges, and logistical difficulties like long waiting times [78,79,80].

## 4. Discussion

The introduction of new vaccines across MICs in the MENA region has been uneven, with some countries demonstrating significant progress while others face delays. For example, Morocco has successfully introduced all three target vaccines (HPV, PCV, and RV), whereas Egypt has yet to integrate any into its National Immunisation Program. Challenges include economic pressures, large refugee populations, and systemic barriers to vaccine access, such as vaccine hesitancy and gender-related barriers, which have direct implications for new vaccine introduction and coverage. These factors, coupled with financing constraints and logistical hurdles, shape the broader landscape of immunisation efforts in the region. The following section explores the key barriers and facilitators to new vaccine introduction, building on these contextual challenges (Table 7).

### 4.1. Barriers to New Vaccine Introduction

A major barrier identified is the lack of recent and comparable epidemiological data. In countries like Algeria, the absence of rotavirus prevalence data makes it difficult to assess the burden of disease, while outdated data in other countries limit the capacity to make informed decisions about new vaccine introductions. This lack of robust data constrains policy decisions and hampers the ability to justify new vaccine introductions based on the disease burden.

Regulatory barriers, such as limited NRA maturity, along with insufficient strategies, investment, innovation capacity, and partnerships, are constraining vaccine manufacturing capacity in the MENA region [26,62,64]. As a result, countries remain heavily reliant on imports, particularly for new vaccines, highlighting significant technical and political challenges in achieving self-sufficiency.

Political and economic instability further complicates vaccine introduction. Egypt’s strained economy and Jordan’s high public debt pose significant challenges to vaccine financing [66,81]. The influx of refugees and internally displaced populations, largely driven by conflict in the region, strains healthcare systems, while economic instability and fragmented service delivery further impede equitable vaccine access [73,78]. Individual-level barriers such as financial constraints—especially in MICs where healthcare funding is largely domestic—language barriers, and caregivers’ lack of knowledge about vaccination schedules exacerbate these challenges [78].

Low public awareness, particularly for HPV, remains another barrier. In most countries, studies indicate low levels of awareness as well as vaccine hesitancy about cervical cancer and the HPV vaccine amongst the public and healthcare providers [45,46,48,50]. In addition, the planned HPV vaccination program in Tunisia is targeted at children in school, with no details on reaching out of school children [41]. Morocco, despite successfully integrating all three new vaccines into its National Immunisation Program, faces challenges with low uptake of HPV, potentially driven by vaccine hesitancy [82]. Gender-related barriers, particularly male-dominated household decision-making and the over-reliance on under-supported female frontline workers, further suppress HPV-vaccine uptake. Evidence from an East-Asian study shows that mothers and grandmothers hesitate to authorise vaccination without a father’s approval, female health workers face harassment and lack training or time to counter misinformation, and clinic hours and facilities seldom match women’s needs—conditions that suggest similar, yet still undocumented, constraints in MENA settings [83].

### 4.2. Facilitators of New Vaccine Introduction

Several facilitators were identified that could promote new vaccine introductions, such as political commitment, availability of support through Gavi MICs and UNICEF, and the potential for local vaccine manufacturing.

The role of local manufacturing could accelerate new vaccine introductions in the future if robust strategies are implemented and appropriate investments are made. Egypt’s Vacsera and Morocco and Algeria’s Institut Pasteur have the potential to reduce dependence on imports and ensure local availability of critical vaccines [62]. Additionally, private sector involvement in countries like Lebanon, where a significant proportion of vaccinations are administered through private providers, highlights potential for public–private partnerships to strengthen immunisation efforts [30].

The availability of Gavi MICs support offers financial and technical assistance to eligible MICs [3]. Support from partners, including UNICEF and WHO, can help countries in the MENA region apply for Gavi MICs funding to help overcome financial and supply barriers. For example, Tunisia’s ability to access catalytic funding from Gavi will provide critical resources to cover the initial costs of HPV procurement and rollout [26].

Beyond supply constraints, targeted demand-generation strategies, such as school-based HPV delivery with opt-out consent, SMS reminders for infant vaccines, and outreach by community health workers, may help improve uptake in MICs according to some studies [84,85,86,87]. Other experts report that trust grows with clear, context-specific messaging (e.g., HPV prevents cancer), transparent reporting of adverse events, rapid misinformation response, and involvement of trusted leaders, especially where gender norms affect consent [88,89]. Aligning these approaches with financing and supply-side support may further accelerate vaccine uptake in MENA.

**Table 7 vaccines-13-00860-t007:** Summary of barriers and facilitators to vaccine introduction.

Theme	Barriers	Facilitators
Epidemiology	Lack of recent and comparable data on disease prevalence. Outdated data limits informed decision-making on NVI	Evidence from high-burden studies (where available) can support NVI; Potential for enhanced surveillance and data collection initiatives
Healthcare infrastructure	Impact of conflict and displacement on healthcare infrastructure in the region disrupts vaccine delivery	Successful infrastructure, like Morocco’s NVI model, offers a scalable approach for strengthening vaccine rollout
Financial resources	Fiscal constraints due to economic vulnerability and public debt	Gavi MICs support with funding applications
Political economy	Political and economic uncertainty; High rates of displacement within and across borders driven by war and conflict	Political commitment to facilitate NVI
Local manufacturing	Limited local production in the region and high dependency on imports	Investing in local manufacturers may reduce dependency on imports and enhance local vaccine accessibility in the future
Awareness and trust	Low public awareness and acceptability of the HPV vaccine; Gender-related barriers	Potential to boost public confidence and enhance uptake through targeted interventions

NVI: New vaccine introductions.

### 4.3. Evidence Gaps and Study Limitations

This study identifies several evidence gaps, particularly the lack of recent epidemiological data. For example, epidemiological data on rotavirus are not available from a comparable, standardised source; thus, all estimates for the MENA region are from separate publications with different sample sizes and data collection methods. It is also unclear whether vaccination coverage estimates include displaced populations. Fragmented procurement mechanisms and limited availability of information on Gavi MICs applications, the changing status of Gavi MICs support eligibility (e.g., as of 2025, Algeria and Iran have moved into upper middle-income status) [32], NITAG recommendations, and long-term immunisation planning make it difficult to evaluate the role of financial and scientific support driving new vaccine introductions. Furthermore, it was difficult to ascertain reliable information on the ‘political economy’ and ‘vaccine regulation and procurement’ domains of the vaccine readiness framework. During expert consultations, it was advised that in-depth, stakeholder engagement is required to obtain these data.

### 4.4. Practical Recommendations and Areas for Further Research

Further research applying the vaccine readiness framework should prioritise gathering local epidemiological data and using in-depth stakeholder engagement across all domains, but especially ‘political economy’ and vaccine regulation and procurement’. The vaccine readiness framework could also be used to benchmark regional progress and identify actionable gaps in political commitment and regulatory preparedness. In addition, the following practical steps are also recommended:Strengthen NITAGs: Establishing and empowering NITAGs in each country could improve evidence-based decision-making and streamline vaccine introduction processes.Enhance local vaccine production: Prioritising investments in local manufacturing capabilities may reduce dependency on imports and improve vaccine accessibility. Collaborations with international partners such as UNICEF and WHO can provide technical support and funding to support these initiatives.Address vaccine hesitancy: Developing culturally sensitive awareness campaigns targeting communities and healthcare providers could combat misinformation and build public trust, particularly regarding HPV vaccines.Optimise use of Gavi MICs support: Eligible countries are encouraged to apply for Gavi MICs funding to bridge initial financial gaps or for technical support, particularly with regard to the introduction of HPV vaccines.Expand coverage to vulnerable populations: Immunisation programmes can explicitly target displaced populations, out-of-school children, and marginalised groups.

## 5. Conclusions

This study evaluated new vaccine introductions in seven MENA MICs: Algeria, Egypt, Iran, Jordan, Lebanon, Morocco, and Tunisia. It assessed the introduction status of HPV, PCV, and RV vaccines, identifying barriers and facilitators. The uneven rollout of HPV, PCV, and RV vaccines in MENA MICs presents a critical opportunity to address preventable diseases through targeted actions. Addressing fiscal constraints, improving local manufacturing, addressing gender barriers, and fostering public trust can enable these countries to make significant strides towards achieving equitable vaccine access. Looking forward, a collaborative, region-wide approach involving governments, international organisations, and local communities is crucial. Together, these efforts can bridge gaps, overcome barriers, and ensure that no child or community is left behind in the fight against preventable diseases.

## Figures and Tables

**Figure 1 vaccines-13-00860-f001:**
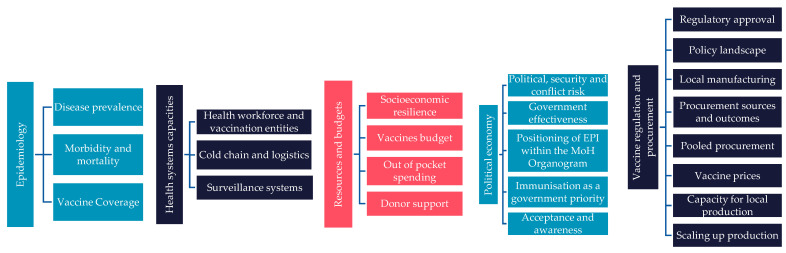
Vaccine readiness framework.

**Figure 2 vaccines-13-00860-f002:**
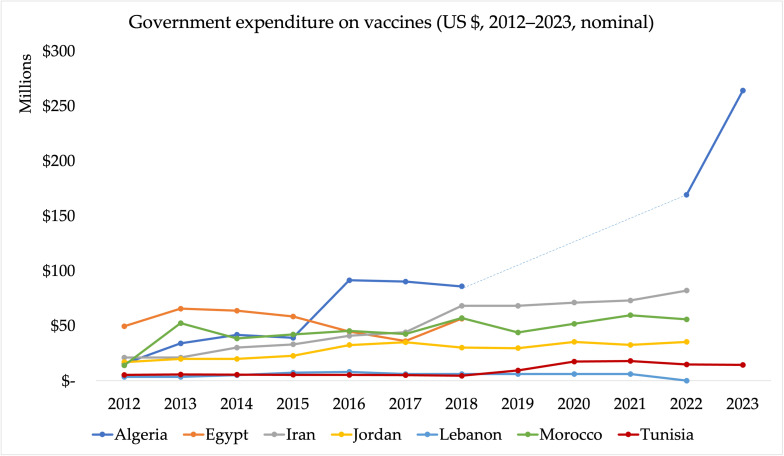
Government expenditure on vaccines (GEV) used in routine immunisation (USD 2012–2022, nominal) (Source: WHO/UNICEF Joint Reporting Forms). Note: The Joint Reporting Form data on government spending on vaccines do not include spending on vaccines used during campaigns or supplementary immunisation activities. The dotted segment represents years for which data were unavailable from the WHO/UNICEF Joint Reporting Form (JRF); values were interpolated or extrapolated from adjacent years for continuity of presentation.

**Table 5 vaccines-13-00860-t005:** Per capita spent on immunisation.

Country	Spend Per Capita [58]
Algeria	USD 2.07/capita (2018)
Egypt	USD 0.58/capita (2018)
Iran	USD 0.82/capita (2019)
Jordan	USD 2.93/capita (2019)
Lebanon	Not available
Morocco	USD 1.20/capita (2019)
Tunisia	USD 0.45/capita (2017)
LMIC medians	USD 0.55 (2011) and USD 1.01 (2014) for LMICs

## Data Availability

The data presented in this study are available in this article.

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
