# Peer review of "New Vaccine Introduction in Middle-Income Countries Across the Middle East and North Africa—Progress and Challenges"

_vaccines, 2025, doi:10.3390/vaccines13080860_

Round 1
Reviewer 1 Report
Comments and Suggestions for Authors
N/A
Comments on the Quality of English LanguageI suggest that you also address and discuss any evidence-based interventions that can be used to promote the awareness of these three vaccines, as well as those that can be used to address vaccine hesitancy.
Author Response
Comments 1: I suggest that you also address and discuss any evidence-based interventions that can be used to promote the awareness of these three vaccines, as well as those that can be used to address vaccine hesitancy.
|
Response 1: Thank you for pointing this out. We agree with this comment. Therefore, we have incorporated some light discussion of evidence-based interventions to promote awareness and address vaccine hesitancy for HPV, rotavirus, and PCV. This is kept light in this manuscript as we are exploring a range of interventions to improve uptake in the region with UNICEF social and behavioral change colleagues, which will feature in a subsequent publications (hopefully).
Please see the adjusted paragraph on page 17, lines 473 – 480.
|
Reviewer 2 Report
Comments and Suggestions for Authors
It was a pleasure to be appointed to review the review article entitled “New Vaccine Introduction in Middle Income Countries across the Middle East and North Africa – progress and challenges” by Chrissy Bishop and other co-authors.
This review article is nicely constructed and very informative, offering a valuable and well-structured detailed discussion on New Vaccine Introduction in Middle Income Countries across the Middle East and North Africa – progress and challenges in preventing vaccine-preventable diseases. The review article is efficiently synthesizes difficult material, offering useful insights and a solid basis for comprehending the subject matter. The depth and clarity of the content render it beneficial for anybody aiming to achieve a comprehensive grasp of this domain. I would congratulate the authors for wonderful piece of activity.
However, I have some comments/suggestion that may be incorporated.
- In an abstract line no. 29 “one country” what what author want to say here is not clear.
- In a material and method section, line no 103, which data extraction tool is used?
- In a results section, line no. 146, does author mean under the five year of age that may be written clearly for the ease of readers understanding.
- Some of the short cuts of term used through the text may be given in full form; the full form may be help readers to understand.
- Figure no. 2, is the data/information not available for Tunisia.
- In the discussion section, suddenly, instead of “other challenges”, may be started like, challenges include…….
Author Response
Comments 1: In an abstract line no. 29 “one country” what author want to say here is not clear
Response 1: Agree. We have edited accordingly, by changing the language to “a single country introduced HPV” and also added “at the time of writing” as since starting this manuscript Tunisia has also introduced HPV. These edits can be found on page 1, lines 29-31.
Comment 2: In a material and method section, line no 103, which data extraction tool is used?
Response 2: Thank you for pointing this out. We created a unique data extraction tool for the purposes of this review. It was created in conjunction with UNCIEF MENARO, to ensure we were collecting data important for their ongoing work packages. To make this clear we have added the term “bespoke” into this paragraph, page 3, paragraph 2, line 105.
Comment 3: In a results section, line no. 146, does author mean under the five year of age that may be written clearly for the ease of readers understanding.
Response 3: Thank you for pointing this out. Exactly that is what we mean. We have edited in this instance and anywhere else in the text to improve clarity. First edit in response to your comment can be seen on page 4, paragraph 3, line 149.
Comment 4: Some of the short cuts of term used through the text may be given in full form; the full form may be help readers to understand.
Response 4: Thank you. We agree with this suggestion. We have retained widely recognised acronyms such as HPV, PCV, RV, and NITAG, but have provided the full forms for lesser-known abbreviations, namely NVI (new vaccine introduction), NIP (National Immunisation Programme), MoH (Ministry of Health), and VRF (Vaccine Readiness Framework). These have been introduced in full at their first mention and clarified in a footnote within the relevant table for clarity and space efficiency. Revisions have been marked in track changes in various places throughout the manuscript, for example page 1, line 110.
Comment 5: Figure no. 2, is the data/information not available for Tunisia.
Response 5: Thank you for this observation. The data for Tunisia is indeed included in Figure 2. However, it may have been overlooked due to earlier colour choices. To improve visibility, we have changed the colour for Tunisia to deep red in the revised figure. The graph (Figure 2) with edited colours is on page 12, line 310.
Comment 6: In the discussion section, suddenly, instead of “other challenges”, may be started like, challenges include…….
Response 6: Thank you, this is a very good point. We have edited so the sentence now starts “Challenges include”. This can be found on page 16, last paragraph line 416.
Reviewer 3 Report
Comments and Suggestions for Authors
Thank you for the opportunity to review this manuscript. In this paper, the authors describe challenges with the introduction of new vaccines to middle-income countries in the Middle East and North Africa (MENA) region, specifically vaccines against HPV, pneumococcal disease, and rotavirus.
This is an interesting and well-written paper. The quality of written English is excellent. (If my own trainees, who are largely native English-speakers, could write as well, I would be delighted.) My significant observation is that it would be interesting, if available, to include a discussion of the impact of vaccine rollouts in the countries that have implemented them successfully. For example, given that Morocco has implemented all three vaccines of interest, how has this affected the incidence of rotavirus and pneumococcal infection in these countries? (I suspect it will take longer to appreciate the impact of HPV vaccination.) Some degree of discussion of this topic would be of interest. Otherwise, this is good work.
Specific comments are as follows:
Lines 148-149 – I think there is confusion here between incidence and prevalence; I doubt that 66% of children under 5 years of age in Lebanon have rotavirus diarrhea at any given moment in time. Recommend changing “prevalence” to “incidence” and including the time period in question (e.g., 66% of children under 5 years of age will experience rotavirus diarrhea during a given 12 month period, at any time during their first 5 years of life, etc.).
Lines 160-161 – Similar to above, I assume this is the lifetime incidence of cervical cancer?
Table 1 – Once again, “prevalence/incidence” are not interchangeable terms. If incidence is reported, it must included a time interval (which may be for the person’s entire lifetime but could also be per year, for example).
Line 204 – I assume this section refers to pediatric PCV usage, not in older and high-risk adults? If it is one of the other, I would specify. If it is both, then I would clarify differences in vaccine uptake between the two groups.
Author Response
Comment 1: Lines 148-149 – I think there is confusion here between incidence and prevalence; I doubt that 66% of children under 5 years of age in Lebanon have rotavirus diarrhea at any given moment in time. Recommend changing “prevalence” to “incidence” and including the time period in question (e.g., 66% of children under 5 years of age will experience rotavirus diarrhea during a given 12 month period, at any time during their first 5 years of life, etc.).
Response 1: Thanks so much for this very helpful comment. Indeed, when we went back to the paper to check the 66% figure it is in fact period prevalence of rotavirus cases out of a total of rotavirus and enteric adenovirus infections, not the number of cases admitted in total with a diagnosis of acute gastroenteritis (AGE). We changed the period prevalence figure to 17%, which is the total number of rotavirus positive cases out of all hospitalized patients admitted with AGE. We also added in the type of data reported (period prevalence in a hospital setting), and added the periods of data collection for each country as a footnote. These edits can be found in Table 1, page 6, line 174. We added further clarification in the text explaining the table, adding “Table 1 summarises country-specific estimates for rotavirus-positive acute gastroenteritis, pneumonia and severe pneumonia in children, and cervical cancer in women. For rotavirus, only period prevalence data among children admitted to hospital with acute gastroenteritis was available from various hospital surveillance studies, no age-standardised data was found.” This can be found on page 4, 3rd paragraph lines 147-152.
Comment 2: Lines 160-161 – Similar to above, I assume this is the lifetime incidence of cervical cancer?
Response 2: Thank you for the detailed review of this data. It is not lifetime incidence; it is one year incidence. The reference reports the incidence as a rate per 100,000 women per year, but we converted this to a % for consistency purposes in the table (all other epidemiological data was reported as a %). To make sure this is clear, we have added the rate and % into the table, and added “one year” into the table description as well as the year of reporting (2020). This is in Table 1, page 6, line 174. In addition, we added some clarification to the paragraph before the table and added “this source reports age standardised cervical cancer incidence for year 2020” on page 4, last paragraph, line 169-170.
Comment 3: Table 1 – Once again, “prevalence/incidence” are not interchangeable terms. If incidence is reported, it must included a time interval (which may be for the person’s entire lifetime but could also be per year, for example).
Response 3: Thank you for pointing this out. We agree the use of “/” is confusing. We have changed the title of the table to “Burden of vaccine preventable diseases” as we do report prevalence for rotavirus and incidence for pneumonia and cervical cancer. This edit can be found on line 174 above the table. We have also added the time interval as a footnote on line 175. Thanks this is an important addition.
Comment 4: Line 204 – I assume this section refers to pediatric PCV usage, not in older and high-risk adults? If it is one of the other, I would specify. If it is both, then I would clarify differences in vaccine uptake between the two groups.
Response 4: Thank you for reminding us of this important clarification. All of our work in this paper refers to children under five years of age, but you are correct to suggest this could be confusing. Especially given the PCV vaccine is recommended to older age groups too. We have added on page 8, line 217 “uptake in children under 5 years of age”.